# Indirect Mechanisms of Transcription Factor-Mediated Gene Regulation during Cell Fate Changes

*Michael R. Larcombe, Sheng Hsu, Jose M. Polo, and Anja S. Knaupp\**

Transcription factors (TFs) are the master regulators of cellular identity, capable of driving cell fate transitions including differentiations, reprogramming, and transdifferentiations. Pioneer TFs recognize partial motifs exposed on nucleosomal DNA, allowing for TF-mediated activation of repressed chromatin. Moreover, there is evidence suggesting that certain TFs can repress actively expressed genes either directly through interactions with accessible regulatory elements or indirectly through mechanisms that impact the expression, activity, or localization of other regulatory factors. Recent evidence suggests that during reprogramming, the reprogramming TFs initiate opening of chromatin regions rich in somatic TF motifs that are inaccessible in the initial and final cellular states. It is postulated that analogous to a sponge, these transiently accessible regions "soak up" somatic TFs, hence lowering the initial barriers to cell fate changes. This indirect TF-mediated gene regulation event, which is aptly named the "sponge effect," may play an essential role in the silencing of the somatic transcriptional network during different cellular conversions.

M. R. Larcombe, S. Hsu, J. M. Polo, A. S. Knaupp
Department of Anatomy and Developmental Biology
Monash University
Clayton, Victoria 3168, Australia
E-mail: anja.knaupp@monash.edu

M. R. Larcombe, S. Hsu, J. M. Polo, A. S. Knaupp
Development and Stem Cells Program
Monash Biomedicine Discovery Institute
Clayton, Victoria 3168, Australia

M. R. Larcombe, S. Hsu, J. M. Polo, A. S. Knaupp
Australian Regenerative Medicine Institute
Monash University
Clayton, Victoria 3168, Australia

J. M. Polo
South Australian Immunogenomics Cancer Institute, Faculty of Health and Medical Sciences
University of Adelaide
Adelaide, South Australia 5005, Australia

J. M. Polo
Adelaide Centre for Epigenetics, Faculty of Health and Medical Sciences
University of Adelaide
Adelaide, South Australia 5005, Australia

## 1. Introduction

During embryonic development and adult homeostasis, the induction of specific transcription factors (TFs) in stem or progenitor cells drives the specification into the diverse cell types that exist in the mature organism.[1] TFs are proteins with DNA-binding domains that recognize specific sequences in the genome usually located in regulatory elements (REs) such as promoters and enhancers. It has been estimated that there are more than 900,000 putative enhancers/REs in the human genome and ≈2,000 TFs.[2–4] TFs act either as activators or as repressors and up-regulate or down-regulate gene expression, respectively.[1,4] To exert their function, TFs usually engage other regulatory proteins, including different TFs, histone modifiers, and chromatin remodelers. In the case of transcriptional activators, this includes recruitment of the basal transcriptional machinery and RNA polymerases (**Figure 1**).[5] It is this intricate interplay between specific TFs, distinct cofactors, and selected REs that regulates the expression of a certain set of genes, which ultimately dictates the function and identity of a cell.

Changes to these core regulatory network interactions such as a shift in TF stoichiometry and/or TF combinations can lead to a switch in cellular identity. For example, during development certain cues such as growth hormones and cytokines induce the expression of specific TFs in a spatiotemporal manner, which in turn mediate the cell fate conversions required for the establishment of the human body. In a pathological context, deregulation of different TFs can lead to the formation of various cancers,[6] further highlighting the central role of TFs in cellular identity control. Consequently, it is not surprising that the use of TFs, either directly (e.g., via TF overexpression) or indirectly (e.g., via external stimuli to activate endogenous TFs), is key to various in vitro cellular conversions such as stem cell differentiations[7,8] and somatic cell reprogramming into induced pluripotent stem cells (iPSCs).[9] Insight into the underlying mechanisms through which TFs mediate changes in cell identity is therefore essential for our understanding of human development and tumorigenesis and hence for the development of new or improved in vitro stem cell differentiation protocols as well as novel anticancer strategies.

While the molecular mechanisms driving most cell conversions remain largely elusive, reprogramming into iPSCs has been studied extensively and has shed light on potential TF

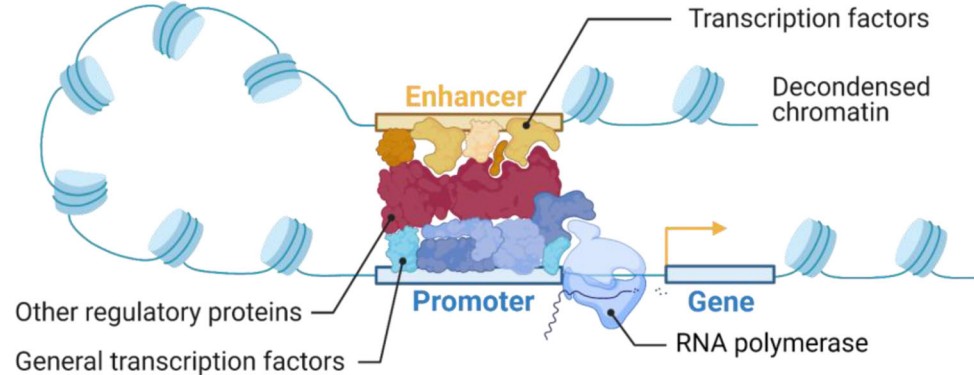

**Figure 1.** A simplified model of transcriptional regulation illustrating the proximal promoter and distal enhancer sequence engaged by TFs and other regulatory proteins. The formation of such a complex at a gene's REs influences the recruitment of RNA polymerase to transcribe the associated gene. Created with BioRender.com.

mechanisms, which might underlie other cellular conversions.[10–13] In this perspective, we discuss the nature of pioneer TFs, the proposed mechanistic models behind direct and indirect TF-mediated cellular identity conversions, and how these regulatory mechanisms could potentially be exploited to further enhance in vitro cell fate changes. Furthermore, we discuss how TF-mediated decondensation of motif-rich "sponge" chromatin can directly sequester TFs away from gene REs to expedite the extinction of established TF networks. We theorize that through this process, which we coined the "sponge effect," cell fate altering TFs initiate redistribution of TFs away from active REs to promote silencing of the initial cell identity network.

## 2. Gene Regulatory Networks and Cellular Identity

The human body is composed of trillions of cells representing hundreds of different cell types with specialized functions.[14,15] Each cell type is characterized by a specific set of active REs and therefore a distinct chromatin accessibility profile that governs a cell-type specific gene expression program.[16] Despite sharing a common transcriptomic signature, cells of the same cell type usually display considerable cell-to-cell variation.[17] This cellular heterogeneity is the product of various concurrent factors including the microenvironment, cell cycle stage, spatial context, external stimuli, and stochastic gene expression.[17] Aside from these continually shifting transcriptional states within a cell type, cells transitioning into a different stable cell type may undergo various transient cellular identity changes en route (e.g., differentiating stem cells).[17] All of these transient and relatively stable cellular states are ultimately the result of specific gene expression programs that are mostly controlled by a certain set of TFs.

While the human genome encodes around 2,000 TFs, it has been proposed that usually five to seven TFs orchestrate the core transcriptional circuitry that determines a cell's identity.[18,19] This concept underpins and becomes especially clear in reprogramming experiments where sets of specific TFs are overexpressed in a given cell type to impose other identities. These master regulators, which interact with most active REs including their own, control the genes that govern cellular identity and are therefore considered to be at the top of the regulatory hierarchy.[20–22] Many of these master regulators are essential

for both the establishment and maintenance of specific cellular states. For example, SC1 is required for the development and maintenance of hematopoietic stem cells,[23,24] PAX7 for satellite cells,[25,26] HNF4α for hepatocytes,[27,28] PAX5[29,30] and EBF1 for B cells,[31,32] TLX for neural stem cells[33,34] and NKX6.1 for pancreatic β-cells.[35] Notably, dysregulation of the master regulators of gene networks is commonly associated with the adoption of pathological cellular identities like those observed in cancer.[36] For example, uncontrolled activation of nuclear factor-kB (NF-κB) blocks apoptosis, promotes cell cycle progression, and limits innate and adaptive immune surveillance mechanisms in several cancers.[37–40] Similarly, reactivation of the TFs OCT4, SOX2, and/or NANOG has been linked to various cancers in which these TFs facilitate numerous malignant aspects including cancer stemness, proliferation and drug resistance.[41–43] Notably, OCT4, SOX2, and NANOG are key to the pluripotent state of embryonic stem cells (ESCs) and iPSCs, enabling unlimited self-renewal as well as the capacity to give rise to all cell types of the mature body.[44]

## 3. The Role of TFs during Forced Cell Fate Changes

While reprogramming into iPSCs using the Yamanaka TFs OCT4, SOX2, KLF4 and C-MYC is arguably the most prominent in vitro cellular conversion due to the unprecedented therapeutic potential of iPSCs, numerous other forced inter-cell conversions have been described. Pioneering work by Sir John Gurdon in 1962 demonstrated the potential of nuclear transfer as a means of changing cell identity and cloning.[45] In 1987, Davis and colleagues made another major scientific discovery in the reprogramming field by showing that fibroblasts can be converted into myoblasts via forced expression of the TF *Myod*.[46] This process of directly transforming one cell type into another without traversing through a pluripotent state is commonly known as transdifferentiation and presents an alternative approach to ESC/iPSC differentiation for the production of clinically relevant cell types. Many other transdifferentiation reactions have since been described including the direct conversion of fibroblasts into neurons via ASCL1, BRN2, and MYT1L, into cardiomyocytes via GATA4, MEF2C, and TBX5 and into hepatocytes using HNF1A, HNF4A, HNF6, ATF5, PROX1 and CEBPA (**Figure 2**).[48–50]

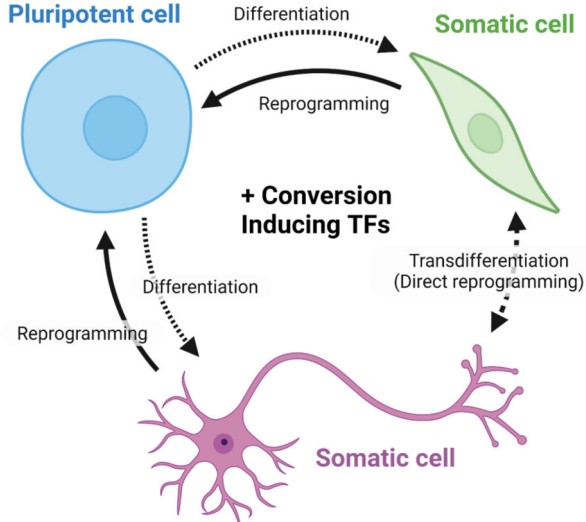

**Figure 2.** Cell fate plasticity represented by the depiction of common in vitro cellular conversions. Pluripotent stem and progenitor cells have the developmental potential to differentiate into mature cell states. These differentiation processes are regulated by TFs in vivo and can be reproduced or enhanced by inducing these TFs in vitro. Somatic cells can be reprogrammed back into a pluripotent state or transdifferentiated directly into an alternative somatic state bypassing a pluripotent state via the use of different TFs. Created with BioRender.com.

Furthermore, TFs can be applied to "enhance" or "forward" differentiate ESCs/iPSCs into various cell types such as neurons via overexpression of *Ascl1* or *Ngn2*, myocytes via *Myod1* or *Esx1* and hepatocytes via *Gata2* or *Gata3*.[7,8,46,47]

In principle, all of these cellular conversions (e.g., reprogramming, differentiation, transdifferentiation) require the erasure of original chromatin signatures, a shift in gene expression patterns, and the subsequent loss of cellular identity to make way for a new transcriptional program. Perhaps the most explored model for TF-mediated loss of cellular identity is the conversion of mouse embryonic fibroblasts (MEFs) to iPSCs.[10–12,48–50] Reprogramming MEFs into pluripotent stem cells is highly inefficient and the few emerging iPSCs endure multiple waves of transcriptional reconfiguration throughout this transformation.[13,44,51,52] Analysis of chromatin structure and transcriptional activity throughout reprogramming has identified two transcriptional waves correlating to a loss of the somatic network in the early stages of Yamanaka TF induction and the late activation of the pluripotency program.[10,13,48,51,52] Accordingly, the forced expression of pluripotency factors *Esrrb*, *Sall4*, and *Zic3* can enhance the efficiency of iPSC generation by supporting the establishment of the pluripotency network.[53–55] Moreover, elevating the levels of the Yamanaka TFs increases reprogramming efficiency and rescues cells refractory to reprogramming, demonstrating that OCT4, SOX2, KLF4, and C-MYC expression levels can be rate-limiting.[10,13,56–60] Additionally, it has been shown that the repression of somatic factors can aid in dismantling the initial barriers to cellular reprogramming.[61,62] For instance, depletion of *Fosl1*, *C-Jun*, *Runx1*, *Cebpa/b*, *Egr1*, or *Fos* during the earliest stages of MEF reprogramming greatly enhances iPSC generation.[11,12,63,64] This should not be surprising, as the sta-

bility of cell identity is key for an organism and as such illustrates that somatic TFs are decisive conservators of the initial cellular state.[10–12,60,63,64] Indeed, cellular identity loss represents the first major bottleneck for the successful generation of iPSCs and likely any other cellular conversion. Sir John Gurdon has coined this as the "resistance of transcriptional activation during reprogramming."[65] Thus, the molecular events that govern this early transcriptional reconfiguration have received considerable attention.[10–13,48–50,63]

Many of the aforementioned TFs used for cellular conversions including MYOD, NEUROD1, ASCL1, OCT4, SOX2, and KLF4 possess the capacity to access their target sites in a non-permissive chromatin state, indicating that they act as pioneer factors (**Table 1**).[8,56–58,66,67] Unlike most TFs, pioneer TFs can bind to their targets in condensed chromatin and initiate an increase in chromatin accessibility, which in turn enables binding of non-pioneer TFs and other regulators to influence gene expression.[56,68] Numerous pioneer TFs have been linked to the distribution of ubiquitous chromatin remodeling factors toward specific gene regulatory sequences. For example, GATA3, MYOD, FOXD3, and OCT4 can independently recruit the chromatin remodeling factor BRG1 to create an accessible environment at their respective target sites.[69–71] OCT4 interacts with several NuRD and SWI/SNF remodeling factors, recruiting secondary factors like INO80 (SWI/SNF) to mediate local chromatin structure.[72,73] Furthermore, it has been shown that KLF4 recruits the DNA demethylase TET2,[74] FOXD3 recruits histone deacetylase 1 and 2 (HDAC1/2)[75] and SOX2 recruits the NuRD transcriptional repressor components LSD1, HDAC1, and MTA2[76] to remodel chromatin structure. This overrepresentation of pioneer factors amongst the TFs used for various cell conversions highlights their ability to trigger extraordinary changes in cellular identity, especially considering that more of the established cell conversion factors might indeed be undiscovered pioneer TFs. While the molecular mechanisms via which TFs mediate various cell fate changes remain largely elusive, somatic cell reprogramming has been studied extensively and provides an excellent and tractable model to provide insight into the early cellular changes. Consequently, in-depth analysis of this cellular conversion has led to several mechanistic models, which potentially also play a role in other cell conversion types.[10–13]

## 4. Mechanistic Models for TF-Mediated Cellular Conversions

In vitro and in vivo cellular conversions between specific cell types are driven by distinct sets of TFs, yet somatic cell reprogramming into iPSCs can be achieved with the same set of four TFs irrespective of the starting cell type. This begs the question; how do the Yamanaka TFs universally convert any epigenome into that of a pluripotent stem cell? How are these TFs able to suppress all somatic identity transcriptional networks? Furthermore, do OCT4, SOX2, and KLF4 target the same genomic regions irrespective of cell type due to their pioneering capacity? If this is not the case, what determines their binding in different cellular contexts? And at the molecular level, how do these pioneer TFs silence any pre-established transcriptional network before activating the pluripotency circuitry? Considering their universal reprogramming capacity, understanding the mechanisms

**Table 1.** Examples of cellular conversion TFs with pioneering capacity.

| Pioneer TFs | Additional factors | Cell conversion | Cell types | Comments | Refs. |
|---|---|---|---|---|---|
| ASCL1 (MASH1) | BRN2, MYT1L,LHX3, HB9, ISL1 and NGN2 | Transdifferentiation | Fibroblasts to motor neurons | | [77, 78] |
| ASCL1 (MASH1) | BRN2 and MYTL1 | Transdifferentiation | Fibroblasts to glutamatergic neurons | BRN2 and MYTL1 can be omitted if co-cultured with glia | [8, 77, 79] |
| ASCL1 (MASH1) and FOXA2 | BRN2, MYT1L and LMX1A | Transdifferentiation | Fibroblasts to dopaminergic neurons | | [77, 80, 81] |
| ASCL1 (MASH1) | | Differentiation | ESCs to glutamatergic neurons | If co-cultured with glia | [8, 77] |
| ASCL1 (MASH1) | | Transdifferentiation | Astroglia to glutamatergic and GABAergic neurons | | [77, 82] |
| ASCL1 (MASH1) and SOX2 | | Transdifferentiation | Pericytes into GABAergic neurons | | [58, 77] |
| ASCL1 (MASH1) | BRN2 and MYTL1 | Transdifferentiation | Hepatocytes to neurons | | [77, 83] |
| CEBPA and PU.1 | | Transdifferentiation | Fibroblasts to macrophages | | [84–86] |
| CEBPB | | Transdifferentiation | Fibroblasts to adipocytes | | [87] |
| EBF1 | | Differentiation | Progenitor cells to B cells | | [88] |
| FOXA1 (HNF3A) | Androgen receptor | Tumorigenesis | Prostate cancer* | | [89, 90] |
| FOXA1 | Estrogen receptor α | Tumorigenesis | Breast cancer* | | [89, 90] |
| FOXA3 | HNF1B | Transdifferentiation | Fibroblasts to bipotential hepatic stem cells | | [91, 92] |
| FOXA3 | HNF1B | Differentiation | ESCs to hepatocytes | | [92, 93] |
| FOXA3 and GATA4 | HNF1A | Transdifferentiation | Fibroblasts to hepatocytes | Concurrent depletion of *Cdkn2a* | [86, 89, 94] |
| GATA2 | GFI1B, C-FOS and ETV6 | Transdifferentiation | Fibroblasts to hematopoietic progenitors | | [95, 96] |
| GATA4 | MEF2C and TBX5 | Transdifferentiation | Fibroblasts to cardiomyocytes | | [86, 97] |
| GATA4 and ISL1 | | Differentiation | Cardiac progenitors to cardiomyocytes | | [86, 98] |
| HNF4A and FOXA1, FOXA2 or FOXA3 | | Transdifferentiation | Fibroblasts to hepatocytes | | [89, 92, 99, 100] |
| KLF4 | | Tumorigenesis | Osteosarcoma, glioma, squamous cell carcinoma, breast, colorectal, and liver cancer* | KLF4 levels positively correlate with disease progression and/or severity | [58, 101–106] |
| MYOD | | Transdifferentiation | Fibroblasts, chondroblasts, smooth muscle cells, or pigmented retinal epithelial cells to myoblasts | | [46, 107, 108] |
| NEUROD1 | | Differentiation | ESCs to neurons | | [67, 109] |
| NEUROG2 | | Transdifferentiation | Astroglia to glutamatergic neurons | | [110, 111] |
| NEUROG2 | ISL1 and LHX3 | Differentiation | ESC to neurons (spinal motor neuron) | | [110, 112] |
| NEUROG2 | | Differentiation | ESCs/iPSCs to neurons | If co-cultured with glia | [7, 110] |
| OCT4 | LMYC, RUNX2, and OSX | Transdifferentiation | Fibroblasts to osteoblasts | | [57, 113] |
| OCT4 | | Transdifferentiation | Fibroblasts to hematopoietic progenitors | | [57, 114] |
| OCT4, SOX2, and KLF4 | C-MYC | Reprogramming | Fibroblasts to iPSCs | While reprogramming is possible in the absence of any individual Yamanaka TF, albeit at a much lower efficiency, depletion of KLF4 seems the most detrimental. | [9, 56, 57, 86, 115] |

*(Continued)*

**Table 1.** (Continued).

| Pioneer TFs | Additional factors | Cell conversion | Cell types | Comments | Refs. |
|---|---|---|---|---|---|
| OCT4, SOX2, and KLF4 | C-MYC | Transdifferentiation | Fibroblasts to angioblast-like multipotent progenitor cells | Induction of a plastic intermediate state prior to mesodermal conversion via media | [57, 116] |
| OCT4 | | Tumorigenesis | Various cancers including glioblastoma, germ cell tumors, bladder, breast, cervical, and liver cancer* | OCT4 levels positively correlate with disease progression and/or severity | [58, 117–122] |
| OLIG2 | | Differentiation | Oligodendrocyte precursor cells to oligodendrocytes | | [123] |
| OLIG2 | | Tumorigenesis | Diffuse intrinsic pontine glioma* | Neural progenitor cells proposed as cell of origin | [123, 124] |
| PAX6 | | Transdifferentiation | Astroglia to neurons | | [125, 126] |
| PAX7 | TPIT | Differentiation | Pituitary intermediates to melanotropes | | [127] |
| PBX1 and MYOD | | Differentiation | Myoblasts to skeletal muscle | | [128, 129] |
| PBX1 and FOXA1 | ERα | Tumorigenesis | ERα-positive breast cancer | | [128] |
| PU.1/SPI1 | CEBPA | Differentiation | iPSCs to macrophages | | [84, 86] |
| SOX2 | FOXG1 and BRN2 | Transdifferentiation | Fibroblasts to tripotent neural precursor cells | | [57, 130] |
| SOX2 | EWS-FLI-1 fusion protein and miRNA145 | Tumorigenesis | Mesenchymal stem cells to Ewing sarcoma cancer stem cells | | [57, 131] |
| SOX2 | | Tumorigenesis | Various cancers including glioblastoma, breast, colorectal, liver, lung, ovarian, and prostate cancer* | SOX2 levels positively correlate with disease progression and/or severity | [58, 132–137] |

*Cell type of origin unknown or disputed.

of action of the Yamanaka factors may uncover the mechanisms that underlie all cellular conversions.

To determine the molecular mechanisms inherent to iPSC formation, several extensive epigenetic analyses have been conducted.[10–13,48,49] Through the integration of various cell identity determinants including chromatin accessibility, reprogramming factor occupancy, DNA methylation, histone modifications, and gene expression, these studies generally agree on several of the molecular aspects that underlie this cell conversion. Specifically, they all recognize the stage-specific and cooperative binding of reprogramming TFs, the rapid silencing of the somatic transcriptional network, and the gradual activation of the pluripotency network during MEF reprogramming.[10–12] Furthermore, it is generally accepted that C-MYC, which greatly enhances reprogramming kinetics and efficiency, functions via a different mechanism to OCT4, SOX2, and KLF4 and plays a secondary role during this process.[57] It has been reported that C-MYC is a major contributor to the first wave of epigenetic events in the initial stages of reprogramming, in part through the recruitment of the NCoR/SMRT-HDAC3 complex that decommissions the somatic cell program.[13,138,139] Beyond these early events, C-MYC exhibits a dual role in which the continued expression is detrimental to the final wave of reprogramming.[139] However, the role of the other three Yamanaka factors in initiating cellular reprogramming remains contentious.[10–12] A major point of debate concerns OCT4, SOX2, and KLF4 and how these TFs, which are generally regarded as transcriptional activators, silence the somatic transcriptional network. Indeed, different mechanistic models for the Yamanaka TF-mediated loss of initial cellular identity have been proposed (**Figure 3**).[138,140,141] Several studies suggest that the reprogramming factors act as pioneer TFs at the start of reprogramming, predominantly targeting closed chromatin.[56–58] One of the first studies to indicate that OCT4, SOX2, and KLF4 preferentially target closed chromatin at the start of reprogramming was conducted by Soufi et al. who interrogated the initial events of human fibroblast reprogramming.[57] Follow-up work by Soufi and colleagues found that OCT4, SOX2, and KLF4, but not C-MYC exhibited significant nucleosomal DNA binding capacity in vitro. Importantly, the degree of nucleosome targeting in vivo corresponded to the factors' relative ability to bind nucleosomes in vitro.[56,58] Notably, their ability to activate repressed chromatin appears to be conserved across species, as the pioneer functions of OCT4, KLF4, and SOX2 have also been observed during MEF reprogramming.[10,11] Furthermore, Soufi's team demonstrated that the capacity of OCT4 to directly interact with nucleosomes can be decoupled from its ability to bind to naked DNA.[56] This work revealed that stable nucleosome interactions (i.e., OCT4 pioneer function) are essential for both OCT4-mediated activation of repressed chromatin early in reprogramming and for pluripotency maintenance (countering the

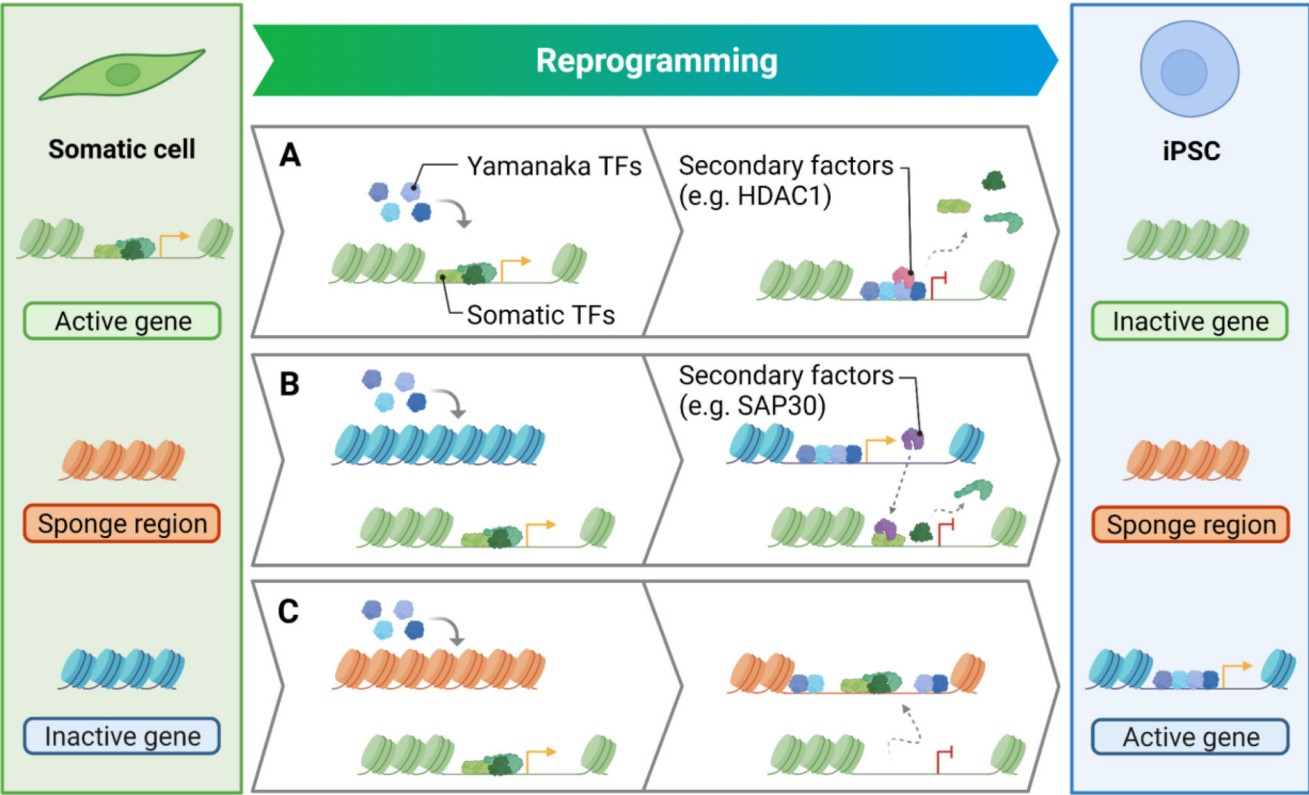

**Figure 3.** Mechanistic models for direct and indirect mechanisms of TF mediated loss of somatic gene expression. A) Epigenomic profiling early in reprogramming revealed extensive redistribution of somatic TFs away from MEF REs coinciding with local accumulation of Yamanaka TFs and subsequent HDAC1 recruitment.[12] In addition to the direct recruitment of epigenetic modifiers, indirect mechanisms have been proposed for the Yamanaka factors' mode of gene suppression.[10,11] B) OCT4, SOX2, and KLF4 induction is suggested to indirectly silence MEF REs through the upregulation of secondary factors like the repressor *Sap30* which in turn engages the REs of crucial somatic TFs.[11] C) We have identified sponge-like chromatin regions rich in Yamanaka and somatic TF motifs, which become transiently accessible upon engagement with OCT4 and SOX2. We propose that these sponge regions are activated transiently by OCT4, SOX2, and KLF4 during reprogramming to induce the redistribution of somatic TFs away from their gene regulatory sites.[10] Created with BioRender.com.

extinction of the pluripotency network).[56] Yet, this contradicts other studies that suggest OCT4, SOX2 and KLF4 preferentially bind to active REs in genes associated with the initial cellular identity.[10–12] The reasons for these observed differences remain largely unclear.

Key differences between these studies are the mouse models utilized and the potentially different levels, combinations (e.g., with or without C-MYC), and stoichiometries of the reprogramming factors.[10–12] Another major difference is the type of reprogramming population analyzed (e.g., bulk reprogramming cultures versus bona fide reprogramming intermediates). As mentioned previously, cellular reprogramming is a very inefficient process with usually less than 3% of somatic cells reaching the pluripotent state.[10,13] Hence, capturing global chromatin and transcriptional signatures from bulk cell populations during reprogramming generates information from a heterogeneous sample that contains refractory fibroblasts. To account for this, cells positioned to reach pluripotency can be enriched in accordance with surface markers restricted to true reprogramming intermediates or through endogenous pluripotency gene expression using fluorescent reporters.[10,11,13]

Notably, Chronis et al., who analyzed bulk populations collected during MEF reprogramming, showed that at the beginning of reprogramming, OCT4, SOX2, and KLF4 predominantly target active (and open) MEF REs.[12] To a lesser extent, these factors also bound to regions that gained accessibility transiently or were permanently accessible during reprogramming.[12] Furthermore, their analysis revealed that approximately half of the enhancers active in MEFs were targeted by OCT4, SOX2, and/or KLF4, binding of which led to depletion of somatic TFs (i.e., FOSL1, CEBPA, CEBPB, and RUNX1) and active chromatin marks early in the reprogramming process.[12] The authors concluded that repression of these particular somatic sites occurs through direct interactions with the reprogramming factors to deny somatic TFs access to their genomic targets hence establishing a transcriptionally repressive environment (Figure 3A).[12] These findings are in agreement with work conducted by Chen and colleagues, who showed that OCT4 preferentially binds to unmethylated and open chromatin regions characterized by active histone marks at the start of MEF reprogramming.[141]

Interestingly, Li et al. integrated the OCT4, SOX2, and KLF4 occupancy data published by Chronis et al. with their own chromatin accessibility data generated from isolated reprogramming intermediates. This analysis also revealed considerable binding of the reprogramming TFs to open REs at the start of reprogramming, which subsequently lost accessibility during this

transition. These regions were largely devoid of Yamanaka consensus motifs, but enriched for somatic TF motifs.[11] However, unlike Chronis et al., Li and colleagues did not attribute silencing of the MEF regulatory network to "promiscuous" Yamanaka TF binding to active MEF REs. Instead, the authors suggest that OCT4, SOX2, and KLF4 target many closed chromatin regions, the majority of which are pluripotency enhancers and promoters, and initiate their opening transiently or permanently.[11] It is this pioneering activity of the reprogramming factors that Li et al. purport to be indirectly responsible for the silencing of the MEF identity network by activating expression of associated genes, such as the corepressor *Sap30*, which in turn eliminates the somatic network (Figure 3B).[11]

Finally, our work, which characterized the same process by extensive epigenomic analysis of purified reprogramming intermediates, found that OCT4 and SOX2 predominantly target regions closed in MEFs and initiate their opening at the beginning of reprogramming.[10] ≈20% of these sites correspond to REs of pluripotency genes that are activated early in the reprogramming process. However, the majority of OCT4 and SOX2 binding events were to regions inaccessible in MEFs and iPSCs.[10] Notably, transient opening of these chromatin regions does not generally correlate with transcriptional activation of the closest genes. Instead, opening of these regions coincides with silencing of the MEF identity network.[10] This suggests that the majority of OCT4 and SOX2 target sites at the start of reprogramming are not cis-REs and most likely have different functions. Whether some of these transient regions act as distal REs remains, however, largely unknown. Interestingly, these transiently accessible chromatin sites are not only enriched with the motifs of OCT4, SOX2, and KLF4 but also of various somatic TFs including members of the AP-1, TEAD, RUNX, and ETS TF families.[10–12] Therefore, we proposed that OCT4, SOX2, and KLF4 engage with closed chromatin regions to open them transiently.[10] This in turn initiates the redistribution of somatic factors to starve fibroblast identity genes of TFs, indirectly forcing the extinction of the somatic transcriptional network (Figure 3C). This molecular "sponge effect" model provides a valid mechanistic explanation for the silencing of the somatic transcriptional network by OCT4, SOX2, and KLF4. Furthermore, it rationalizes how the Yamanaka factors can quash the influence of somatic TFs on chromatin structure and function, favoring the activities of the succeeding factors. Similar to our findings during MEF reprogramming (11,486 peaks, 22.4% total), our analysis of chromatin dynamics during human reprogramming identified a subset of regions (8,411 peaks, 14% total) which experienced a transient increase in chromatin accessibility.[59] Importantly, these transiently accessible sites were also enriched for both somatic and pluripotency TF motifs including FOSL1, JUNB, OCT4, SOX2, and KLF4 suggesting that the sponge effect plays a role during fibroblast reprogramming of different species.[59] The degree to which these "sponge regions" are involved in the silencing of the somatic gene regulatory network during iPSC formation and whether this indirect mechanism of transcriptional regulation extents to other cellular conversions warrants, however, further investigation.

Collectively, these studies suggest that OCT4, SOX2, and KLF4-mediated activation of pluripotency genes occurs predominantly through direct interactions with relevant REs, whereas closure of MEF-specific REs is driven by direct and indirect regula-

tory mechanisms with debated contributions.[10–12] Importantly, all of these studies observe binding of OCT4, SOX2, and KLF4 to genomic regions that are initially closed, open early in reprogramming, and are closed again in the final cellular state. These transiently accessible regions are rich in consensus motifs of the reprogramming factors as well as various somatic TFs including FOSL1 and RUNX1.[10–12] Interestingly, the concept that OCT4, SOX2, and KLF4 induce somatic TF-relocation to these transient regions is strengthened when analyzing purified cell populations, suggesting that this mechanism plays a key role during cellular reprogramming. The extent to which this sponge mechanism influences reprogramming and whether it also underlies reprogramming of other cell types into iPSCs or maintenance of the pluripotent state remain, however, unclear. Cooperative Tbinding is often necessary for TFs to engage with specific regions by concurrently accessing sites that these factors cannot bind independently.[142] Therefore, it is feasible that these sponge-like regions are poised in a near-accessible (OCT4-SOX2 interacting) state in pluripotent stem cells, awaiting cooperative interactions from additional somatic TFs to completely engage. This would effectively work as a buffering mechanism, creating a minimal requirement for somatic TF abundance prior to influencing lineage specific genes for differentiation. Conversely, the sequestration of somatic TFs to sponge-like chromatin regions may not be actively driven by cooperative Yamanaka factor engagement, but may instead be explained by the promiscuous binding of somatic TFs to lower affinity regions. The true extent to which OCT4, SOX2, and KLF4 mediate the opening of these sponge-like regions and whether this effect is universal to all cellular conversions will hopefully be realized through recent advances in the epigenomic field. This includes the development of highly sensitive low input (e.g., single cell) genome-wide profiling techniques as well as novel single locus interrogation approaches.[143–147]

Importantly, TF sequestration may be a universal property of pioneer TFs for driving other in vitro and in vivo cell conversions including transdifferentiations, differentiation, and oncogenesis, hence providing a novel target for manipulation of cell fate as therapeutic strategies and various other applications.

## 5. Exploiting Indirect Gene Regulatory Mechanisms

In order to maintain finely tuned gene regulatory networks, complex mechanisms exist to ensure that appropriate gene expression is enforced in accordance with cellular demands. For example, OCT4 plays an essential role in pluripotent cells during early embryonic development as well as in in vitro ESC/iPSC maintenance and subtle changes in its abundance will induce differentiation.[148] Thus, TF activities have to be tightly regulated as small changes in TF levels can exert large changes in global gene expression programs.

As discussed in detail above, recent evidencesuggests that OCT4 and SOX2 induce the sequestration of somatic TFs toward transiently accessible sponge-like chromatin early in reprogramming. The degree to which this sponge effect can influence cellular identity and whether similar mechanisms underlying natural cellular conversions is currently, however, unknown. Importantly, TF-sequestration strategies utilizing synthetic double-stranded

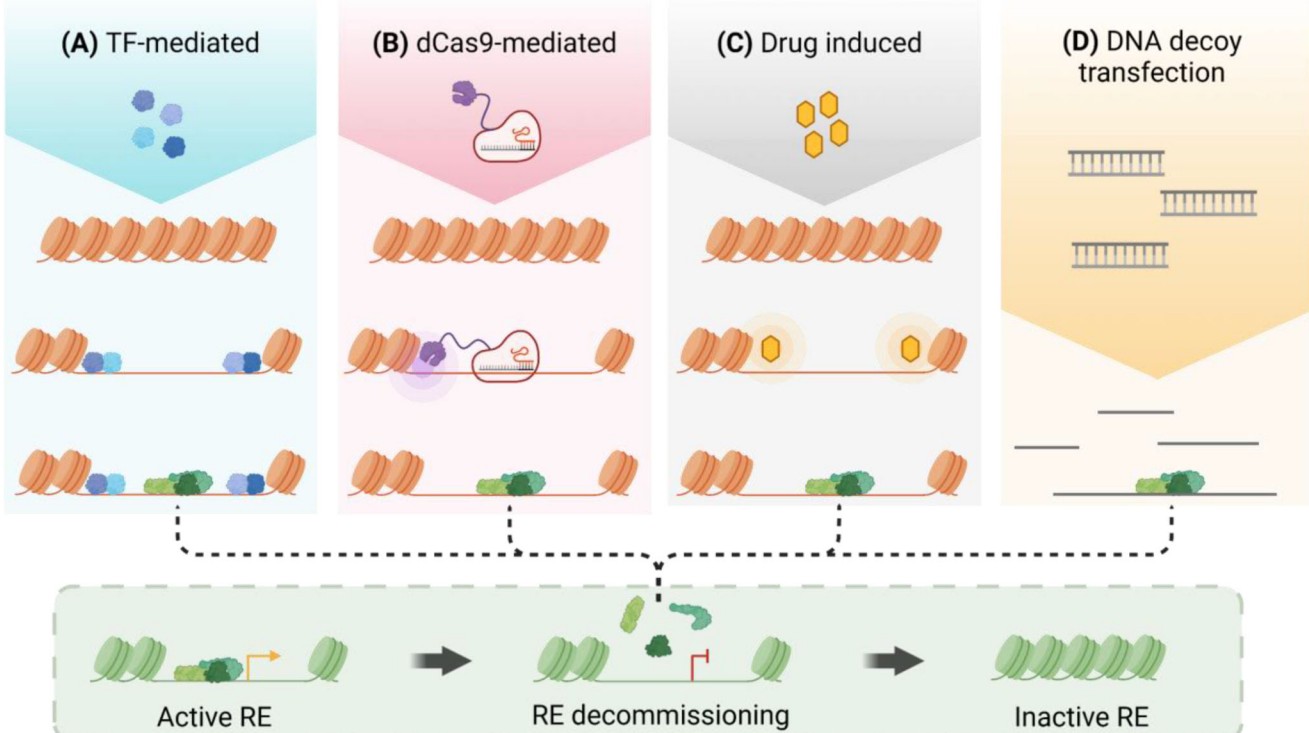

**Figure 4.** Strategies to regulate TF activities via sequestration to chromatin sponge regions or decoy DNAs. Four mechanisms are proposed to deprive an active gene of its regulatory TFs: Targeted activation of sponge-like chromatin containing specific TF binding sites via A) forced expression of pioneer TFs,[10] B) utilization of CRISPR-dCas9-activators,[155] or C) addition of small molecules.[157] Additionally, D) synthetic DNA oligos can be introduced to overwhelm TFs with non-functioning TF binding sites. Created with BioRender.com.

DNA molecules that contain high-specificity TF binding sites have been utilized successfully to manipulate gene regulatory networks in vitro and in vivo (**Figure 4**).[149–152] For example, the administration of such synthetic DNA decoys has significantly diminished the stemness of cancer stem cells by sequesteringoncogenic and pluripotency factors (NF-κB, E2F, and STAT3[151] or SOX2, OCT4 and C-MYC[150]) as well as reduced xenograft tumor volumes in mice treated with intratumoral injections of these bait molecules.[150,151] Furthermore, local administration of NF-κB decoy DNA reduced NF-κB activity and enhanced the antitumor effects of irradiation on subcutaneous human lymphoma xenotransplants in mice.[152] In this study, the combination of irradiation and NF-κB decoy DNA injections, but not independent administration of either was capable of inducing tumor regression and prevented the growth of secondary tumors.[152] Similarly, decoy DNAs containing STAT3 binding sequences reduced excessive production of pro-inflammatory mediators like HMGB1, offering protection against major end-organ injury and improving survival rates of mice following cecal-puncture-induced sepsis.[149] Furthermore, targeted suppression of SP1 activity via sequestration attuned dysregulated extracellular matrix gene expression, reducing the accumulation of excess matrix components like collagen and limitting the severity of tubulointerstitial fibrosis in rats.[153] Altogether, these reports are in agreement with the idea that redistribution of TFs to newly available binding sequences can effectively manipulate existing gene expression networks. Interestingly, a study using a synthetic system in budding yeast suggests that decoy sites may not simply serve as competitive inhibitors but can also transform the regulatory graded dose–response of a positive-feedback loop into a bimodal (switch-like) reaction.[154] The authors of this work suggest that this provides a simple mechanism to generate an ultrasensitive regulatory response necessary for dynamic biological networks.[154]

Whether dynamically activated sponge-like heterochromatin exists as an integral component of the complicated gene regulatory framework remains to be determined. However, this indirect gene regulatory mechanism provides a fascinating new concept to traditional direct transcriptional control.

Verification of TF sponge domains capable of attuning specific TF activities would permit the targeted manipulation of transcriptional networks to influence cell fate. For example, strategies employing drugs with potent and sequence-specific capacities to decondense chromatin or using dCas9-guided chromatin remodeling systems may activate regions rich in certain TF motifs for direct sequestration of distinct factors.[155,156] In support of this, the drug-induced decondensation of chromatin explicitly at DNA satellite V (GAGAA) repeats increased the global amount of high-affinity GAGA factor (GAF) binding sites leading to the subsequent loss of GAF-related functions in *Drosophila melanogaster*.[157] This reduced availability of GAF to engage with gene regulatory sites caused developmental abnormalities reminiscent of partial GAF depletion, indicating that highly repetitive, non-coding regions can be directly manipulated to influence TF activities and global gene expression. Additionally,

treatment of preadipose embryonic fibroblasts with growth hormone triggers relocalization of CEBPA to major a-satellite DNA in heterochromatin[158] while TF-induced depletion of CEBPA from the heterochromatic compartment coincides with increased CEBPA binding at euchromatic regions and enhanced transcriptional output of CEBPA regulated genes.[159] Together, these experiments indicate that TF sequestration to non-coding repetitive DNA sequences is a manipulable phenomenon capable of controlling TF and hence gene activities. With the development of novel chromatin reorganizing drugs, multiplexed dCas9-guided chromatin remodeling systems, and/or identification of TF relocating molecules, we envisage that sequestration of specific TFs will be commonly utilized to modify TF activities and cellular identity. Importantly, manipulation of these mechanisms should allow for greater control over biological systems in order to enhance the efficiency of cell-fate transitions and correct dysregulated gene expression programs of diseased cells.

## 6. Conclusion

Pioneer TFs recognize full or partial DNA binding motifs exposed on the surface of histones to initiate the reconstruction of silent chromatin in a sequence-specific manner. This feature is unique to pioneer TFs and is paramount for the reorganization of epigenetic signatures that prompt changes in cell identity. As such, cellular conversions via the use of pioneer TFs including the Yamanaka factors OCT4, SOX2, and KLF4, have become increasingly popular for modeling developmental processes, generating therapeutically relevant tissues, and investigating the molecular mechanisms that support cell fate decisions. Our current understanding of the means by which these TFs enforce the loss of cellular identity is limited, with evidence amounting to support both direct and indirect mechanisms of action. For example, there is data suggesting that during fibroblast reprogramming into iPSCs OCT4, SOX2 and KLF4 engage active somatic genes, recruit epigenetic silencers and displace somatic TFs to directly suppress the somatic cell identity network. Additionally, the reprogramming TFs appear to directly upregulate chromatin remodelers and other factors, which then aid in the silencing of the somatic transcriptional network. However, the fact that the Yamanaka factors can reprogram practically any cell type irrespective of its epigenome is evidence that their ability to target closed chromatin plays an essential role in their capacity to initiate cellular reprogramming. Indeed, there is evidence suggesting that OCT4, SOX2, and KLF4 bind extensively to regions in closed chromatin and induce opening of these sites. Importantly, these chromatin regions are rich in somatic TF motifs and we propose that their increased accessibility drives redistribution of the associated somatic factors away from active REs, indirectly silencing the initial cell identity network; a mechanism we have coined a sponge effect. Whilst understanding the sponge effect is still in its infancy, several instances of targeted TF sequestration approaches have successfully attuned TF activities in vitro and in vivo. For example, interventions with synthetic DNA molecules containing high-affinity TF binding motifs have successfully been utilized to manipulate TF activities and to decommission the regulatory networks supporting diseased cells. We foresee that the discovery of sponge-like mechanisms in "natural" cell conversion processes such as differentiation and development will be followed closely

with their targeted manipulation, as is the pattern with other gene regulatory mechanisms (e.g., shRNAs, siRNAs, CRISPR/Cas9).

## Supporting Information

Supporting Information is available from the Wiley Online Library or from the author.

## Acknowledgements

This work was supported by a National Health and Medical Research Council (NHMRC) grant (APP1069830) to J.M.P. J.M.P. was supported by the Australian Research Council (ARC) Stem Cells Australia Special Initiative, an ARC Future Fellowship (FT180100674) and a Silvia and Charles Viertel Senior Medical Research Fellowship. A.S.K. was supported by an NHMRC ECF (APP1092280). J.M.P. and A.S.K. were supported by an ARC Discovery Project (DP210104029). M.R.L. was supported by a Melbourne Biomedicine Discovery Institute Postgraduate Discovery Scholarship. The authors thank Flowcore, the Monash Proteomics and Metabolomics Facility, the MHTP Medical Genomics Facility, Micromon, the Monash Bioinformatics and Histology Platforms. The Australian Regenerative Medicine Institute was supported by grants from the State Government of Victoria and the Australian Government. The figures in this document were created with BioRender.com. After initial online publication, the title of this article was corrected on December 12, 2022, as in the original version the word "Cell" was incorrectly written as "CELL".

## Conflict of Interest

The authors declare no conflict of interest.

## Peer Review

The peer review history for this article is available in the Supporting Information for this article.

## Keywords

cellular reprogramming, decoy DNA, iPSCs, pluripotency, sponge effect, transcription factor, transcription factor sequestration, transiently accessible chromatin regions

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

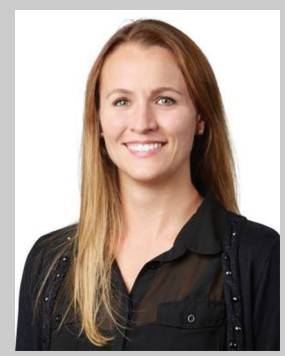

**Anja S. Knaupp** is a senior research fellow, who leads a research team within the Epigenetics and Reprogramming Laboratory in the Department of Anatomy and Developmental Biology, the Australian Regenerative Medicine Institute and the Biomedicine Discovery Institute at Monash University (Melbourne, Australia). Anja is a biochemist by training and received funding from several sources including the Australian Research Council and the Australian National Health and Medical Research Council for her research. Her work centers on understanding the molecular mechanisms that control cellular identity such as transcription factor modes of action and the composition and function of the multimolecular regulatory complexes formed at specific genomic regions.

