## [**Supplementary Information**: Record of Transparent Peer Review · Advanced Genetics]

Indirect mechanisms of transcription factor-mediated gene regulation during cell fate changes

Michael R Larcombe, Sheng Hsu, Jose M Polo and Anja S Knaupp*

* Corresponding author

Date submitted: 05 Apr 2022

Editors: Kerstin Brachhold, Andrew L. Hufton

1st Peer Review Decision

08 Aug 2022

Dear Dr Knaupp,

Thank you for submitting your manuscript entitled "Indirect mechanisms of transcription factor-mediated gene regulation during cell fate changes" (Perspective, No. ggn2.202200015) to Advanced Genetics. The reviewers' reports and comments are included at the end of this e-mail.

I am pleased to inform you that your manuscript has been recommended for publication pending satisfactory revisions. I invite you to respond to the reviewers' comments and make the necessary changes to your manuscript.

When revising your manuscript, please aim for concise wording wherever possible. In agreement with the reviewers, I felt that this Perspective was overall already quite well-written, but I do think there are some places where you may be able to streamline the wording. Please also keep an eye out for long paragraphs and consider breaking them into shorter paragraphs. I think the Introduction, for example, would be a good place where this may be helpful.

Statistics: For original research, please check that your manuscript includes a sub-section entitled "Statistical Analysis" at the end of the Experimental Section that fully describes the following information: 1. Pre-processing of data (e.g., transformation, normalization, evaluation of outliers), 2. Data presentation (e.g., mean \pm SD), 3. Sample size (n) for each statistical analysis, 4. Statistical methods used to assess significant differences with sufficient details (e.g., name of the statistical test including one- or two-sided testing, testing level (i.e., alpha value, P value), if applicable post-hoc test or any alpha adjustment, validity of any assumptions made for the chosen test), 5. Software used for statistical analysis.

Figure legends: Please make sure that all relevant figure legends contain the information on sample size (n), probability (P) value, the specific statistical test for each experiment, data presentation and the meaning of the significance symbol.

A more detailed checklist can be found here: <https://www.advancedsciencenews.com/road-better-data-presentation-dos-donts/>

To submit your revision, go to <https://www.editorialmanager.com/advgenet/> and log in as an Author using your username (*****) and password. Your submission can be found under the menu item "Submissions Needing Revision". The changes to your manuscript should be highlighted in a different color in the primary "Revised Manuscript" file.

Please provide a point-by-point response letter addressed to the reviewers, including a list of changes made and a rebuttal to any comments with which you disagree. You may copy the letter into the "Respond to Reviewers" box (if it is plain text only) or upload it as a "Response Letter to Reviewers" item (if it contains figures, tables, or special formatting such as formulas or equations). If necessary, you may also upload a separate revision cover letter addressed to the editor with any other information not intended for reviewers as a "Cover Letter to Editor" item. You will also be asked to upload a .zip archive containing the production data that will be used if your manuscript is accepted. See below for more details.

We should receive your revised manuscript by 18 Aug 2022. Once we receive your revised manuscript, we will provide a final decision as soon as possible.

Should you need extra time to complete your revision, do not hesitate to contact the editorial office.

Yours sincerely,

Andrew Hufton

--

Andrew Lee Hufton, PhD
Editor-in-Chief, Advanced Genetics
ahufton@wiley.com
Wiley

P.S. Please help avoid delays by referring to the Manuscript Preparation Checklist (<http://www.advgenet.com/authorguidelines>) and use the appropriate article template when preparing your revised manuscript. Please also follow the instructions to prepare and upload your Production Data materials. These include: the full, non-highlighted text of your manuscript (with all figures with a resolution of at least 300 dpi, schemes, and tables) in editable format - Word DOC/X or LaTeX; a short summary (50-60 words) and an eye-catching color image for the Journal's Table of Contents; and, if applicable, Supporting Information*.

*The Supporting Information document(s) will be published alongside your article and should be non-highlighted and ready for publication. Video formats may also be included.

REVIEWER REPORT:

Please note that reviewers may not be numbered consecutively. Where reviewers have provided additional files, these are available here: *****

EVALUATION:

Reviewer's Responses to Questions

Is the topic timely and appropriate for the research community?

Reviewer #1: Yes

Reviewer #3: Yes

Does the manuscript provide a balanced insightful view?

Reviewer #1: Yes

Reviewer #3: Yes

Are the ideas presented reasonable with respect to the supporting literature?

Reviewer #1: Yes

Reviewer #3: Yes

Which aspects of scholarly presentation require improvement (if any)?

Reviewer #1:

*Clarity

Reviewer #3:

*Clarity

COMMENTS TO AUTHOR:

Reviewer #1: I found this perspective article by Larcombe and colleagues well-presented and rather interesting, particularly their hypothesis of pioneer transcription factors enabling the sequestration of somatic transcription factors to enable somatic cell reprogramming. I only have some minor comments:

1-In the Abstract, I am not sure what is meant by direct, indirect, and secondary.

2-In page 4 line 54, I think that somatic cell nuclear transfer is at least as shocking conceptually as somatic cell reprogramming and transdifferentiation by exogenous factors.

3-NFKB is written wrong in multiple places.

4-I would suggest that the authors elaborate more on the previous hypothesis of pioneer transcription factors in somatic cell reprogramming (the 2 Cell papers by Soufi and colleagues).

5-I would also suggest that the authors discuss clearer that the Yamanaka factors can be activators and repressors, and in particular the role of c-MYC in repressing the somatic and pluripotent programs through interaction with NCOR/SMRT co-repressors (Nature Cell Biology 2018 by Zhuang and colleagues).

6-Do the authors think that the sponge effect versus the chromatin remodeling effect of the pioneer transcription factors might be different in mouse and human somatic cell reprogramming?

Reviewer #3: The review manuscript is generally well constructed and provides a thoughtful overview and discussion of the new mechanistic model to address a major gap in the cell reprogramming field; how to extinct pre-existing gene regulatory programs. Overall, this review is valuable scholarly literature, providing insights into cell fate changes in cell reprogramming, development, and tumorigenesis. Some areas need better clarification, as I will outline below.

* What the "sponge effect" represents was not explained in the Abstract and Introduction, even though it is the key mechanism to be discussed in the review. I had no idea about the concept until reading section 4. This needs to be explained earlier.

* It is unclear whether sponge-like regions only function in redistributing somatic TFs in chromatin or also function as cis-regulatory regions for activating target genes during cellular reprogramming.

--

Dr Andrew Hufton, Editor

Advanced Genetics

E-mail: AdvGenet@wiley.com

Tel: +49(0)6201-606-362

<http://www.advgenet.com>

Authors' Response to 1st Review

09 Sep 2022

We would like to thank both reviewers for their time and valuable input. We have now addressed all of their comments and we believe that the manuscript is now of greater quality. Please find the response to all the points raised by the reviewers below:

Reviewer #1: I found this perspective article by Larcombe and colleagues well-presented and rather interesting, particularly their hypothesis of pioneer transcription factors enabling the sequestration of somatic transcription factors to enable somatic cell reprogramming. I only have some minor comments:

1. In the Abstract, I am not sure what is meant by direct, indirect, and secondary.

Thank you for bringing this to our attention. We realise that this isn't clear enough for an introductory statement and have clarified as follows:

ABSTRACT:

TFs are the master regulators of cellular identity, capable of driving cell fate transitions including differentiations, reprogramming and transdifferentiations. Pioneer TFs recognize partial motifs exposed on nucleosomal DNA, allowing for TF-mediated activation of repressed chromatin. Moreover, there is evidence suggesting that certain TFs can repress actively expressed genes either directly through interactions with accessible regulatory elements or indirectly through mechanisms that impact the expression, activity or localization of other regulatory factors. Recent evidence suggests that during reprogramming, the reprogramming TFs initiate opening of chromatin regions rich in somatic TF motifs that are inaccessible in the initial and final cellular states. We postulate that analogous to a sponge, these transiently accessible regions "soak up" somatic TFs, hence lowering the initial barriers to cell fate changes. This indirect TF-mediated event, which we've aptly named the "sponge effect", may play an essential role in the silencing of the somatic transcriptional network.

2. In page 4 line 54, I think that somatic cell nuclear transfer is at least as shocking conceptually as somatic cell reprogramming and transdifferentiation by exogenous factors.

We thank the reviewer for pointing this out and we agree that nuclear transfer experiments pioneered this field. We have now reworded the entire paragraph to acknowledge Robert Davis's TF-guided reprogramming and the prior discoveries made by John Gurdon's work.

While reprogramming into iPSCs is arguably the most prominent in vitro cellular conversion due to the unprecedented therapeutic potential of iPSCs, numerous other forced inter-cell conversions have been described. Pioneering work by Sir John Gurdon in 1962 demonstrated the potential of nuclear transfer as a means of changing cell identity and cloning. (Gurdon, 1962) In 1987, Davis and colleagues made another major scientific discovery in the field of cellular reprogramming demonstrating that fibroblasts can be converted into myoblasts by forcing the expression of the TF MYOD. (Davis et al., 1987) This process of directly transforming one cell type into another without traversing through a pluripotent state is commonly known as transdifferentiation and presents an alternative approach to ESC/iPSC differentiation for the production of clinically relevant cell types. Many other transdifferentiation reactions have since been described including the direct conversion of fibroblasts into neurons via ASCL1, BRN2 and MYT1L, into cardiomyocytes via GATA4, MEF2C and TBX5 and into hepatocytes using HNF1A, HNF4A, HNF6, ATF5, PROX1 and CEBPA (Figure 2). (Du et al., 2014; Ieda, 2011; Vierbuchen et al., 2010) Furthermore, TFs can also be applied to "enhance" or "forward" differentiate ESCs/iPSCs into various cell types such as neurons via overexpression of ASCL1 or NGN2, myocytes via MYOD1 or ESX1 and hepatocytes via GATA2 or GATA3. (Chanda et al., 2014; Davis et al., 1987; Yamamizu et al., 2013; Zhang et al., 2013)

In principle, all of these cellular conversions (e.g. reprogramming, differentiation, transdifferentiation) require the erasure of original chromatin signatures, a shift in gene expression patterns and the subsequent loss of cellular identity to make way for a new transcriptional program. Perhaps the most explored model for TF-mediated loss of cellular identity is the conversion of mouse embryonic fibroblasts (MEFs) to iPSCs.

3. NFKB is written wrong in multiple places.

We are not exactly sure what is incorrect here. As far as we can see, NF- κ B is the correct written form for this transcriptional complex (as originally defined in 1986 by Ranjan Sen &

David Baltimore's back-to-back Cell papers (Sen & Baltimore, 1986, 2006)). We will alter the manuscript to initially define NF- κ B as nuclear factor- κ B, however, we are happy to change this to what the editor believes is appropriate.

For example, uncontrolled activation of nuclear factor- κ B (NF- κ B) blocks apoptosis, promotes cell cycle progression and limits innate and adaptive immune surveillance in several cancer cell types.

4. I would suggest that the authors elaborate more on the previous hypothesis of pioneer transcription factors in somatic cell reprogramming (the 2 Cell papers by Soufi and colleagues).

We appreciate the reviewers' input and agree that we should elaborate on Soufi et al. 's key findings regarding pioneer TFs and their activities in early reprogramming. Please see the end of the following paragraph in comment 5 as both comments are addressed in the same body of text.

5. I would also suggest that the authors discuss clearer that the Yamanaka factors can be activators and repressors, and in particular the role of c-MYC in repressing the somatic and pluripotent programs through interaction with NCOR/SMRT co-repressors (Nature Cell Biology 2018 by Zhuang and colleagues).

We agree with the reviewer that adding insight into the known roles of C-MYC during somatic cell exit and pluripotency acquisition prior to shifting focus improves the overall flow. We thank the reviewer for this suggestion and have adjusted the manuscript as outlined below.

To determine the molecular mechanisms underlying iPSC formation, several extensive epigenetic analyses have been conducted.(Chronis et al., 2017; Hussein et al., 2014; Knaupp et al., 2017a; Li et al., 2017a; Polo et al., 2012; Schwarz et al., 2018) Through the integration of various cell identity determinants including chromatin accessibility, reprogramming factor occupancy, DNA methylation, histone modifications and gene expression, each of these studies generally agree on several of the molecular aspects that underlie this cell conversion. Specifically, they all recognize the stage-specific and cooperative binding of reprogramming TFs, the rapid silencing of the somatic transcriptional network and the gradual activation of the pluripotency network during MEF reprogramming.(Chronis et al., 2017; Knaupp et al., 2017b; Li et al., 2017b) Furthermore, it is generally accepted that C-MYC, which greatly enhances reprogramming kinetics and efficiency, functions via a different mechanism to OCT4, SOX2 and KLF4 and plays a secondary role during this process.(Soufi et al., 2012) It has been reported that CMYC is a major contributor to the first wave of epigenetic events in the initial stages of reprogramming, in part through the recruitment of the NCoR/SMRT-HDAC3 complex that decommissions the somatic cell program.(Polo et al., 2012; Sridharan et al., 2009; Zhuang et al., 2018) Beyond these early events, C-MYC exhibits a dual role in which the continued expression is detrimental to the final wave of reprogramming.(Zhuang et al., 2018) However, the role of the other three Yamanaka factors in initiating cellular reprogramming remains contentious.(Chronis et al., 2017; Knaupp et al., 2017b; Li et al., 2017b) A major point of debate remains regarding how OCT4, SOX2 and KLF4, which seem to generally act as transcriptional activators, silence the somatic transcriptional network. This has led to the proposal of different mechanistic models for the TF-mediated loss of initial cellular identity (Figure 3).(J. Chen et al., 2016; X. Chen et al., 2008; Sridharan et al., 2009) Several studies have suggested that the reprogramming factors act as pioneer TFs at the start of reprogramming, predominantly targeting closed chromatin.(Roberts et al., 2021; Soufi et al., 2012, 2015) One of the first studies to suggest that OCT4, SOX2 and KLF4

preferentially target closed chromatin at the start of reprogramming was conducted by Soufi et al. who interrogated the initial events of human fibroblast reprogramming.(Soufi et al., 2012) Follow-up work by Soufi and colleagues found that OCT4, SOX2, and KLF4, but not C-MYC exhibited significant nucleosomal DNA binding in vitro. Importantly, the degree of nucleosome targeting in vivo corresponded to the factors' relative ability to bind nucleosomes in vitro.(Soufi et al., 2015) This ability to activate repressed chromatin appears to be conserved across species, as the pioneer functions of OCT4, KLF4 and SOX2 have also been observed during MEF reprogramming. (Knaupp et al., 2017a; Li et al., 2017a) Furthermore, Soufi's team demonstrated that the capacity of OCT4 to directly interact with nucleosomes can be decoupled from its ability to bind to naked DNA.(Roberts et al., 2021) This work revealed that stable nucleosome interactions (e.i. OCT4 pioneer function) are essential for both OCT4mediated activation of repressed chromatin early in reprogramming and for pluripotency maintenance (countering the extinction of the pluripotency network).(Roberts et al., 2021)

6. Do the authors think that the sponge effect versus the chromatin remodelling effect of the pioneer transcription factors might be different in mouse and human somatic cell reprogramming?

This is a very interesting question. Similar to our analysis of MEF reprogramming, our human fibroblast reprogramming analysis published in Nature 2020 identified a subset of regions that show transient accessibility during reprogramming. Complementary to our mouse reprogramming findings, transient regions during human reprogramming are likewise enriched for FOSL1, JUNB, OCT4, SOX2, NANOG and KLF4 binding motifs. We believe that the nature of these transiently accessible regions and the likelihood of sponge-like mechanisms existing in the human system warrant further investigation. Indeed, we comment on this in the original manuscript on page 14, lines 33-34. We have now briefly expanded upon this statement for further clarification.

Similar to our findings during MEF reprogramming (11,486 peaks, 22.4% total), our analysis of chromatin dynamics during human reprogramming identified a subset of regions (8,411 peaks, 14% total) which experienced a transient increase in chromatin accessibility.(Knaupp et al., 2017a; Liu et al., 2020) Importantly, these transiently accessible regions were also enriched for somatic and pluripotency TF motifs including

FOSL1, JUNB, OCT4, SOX2 and KLF4. This suggests that the sponge effect plays a role during fibroblast reprogramming of different species.(Liu et al., 2020) The degree to which these "sponge regions" are involved in the silencing of the somatic gene regulatory network during iPSC formation and whether it extends to other cellular conversions warrants further investigation.

Reviewer #3: The review manuscript is generally well constructed and provides a thoughtful overview and discussion of the new mechanistic model to address a major gap in the cell reprogramming field; how to extinct pre-existing gene regulatory programs. Overall, this review is valuable scholarly literature, providing insights into cell fate changes in cell reprogramming, development, and tumorigenesis. Some areas need better clarification, as I will outline below.

1. What the "sponge effect" represents was not explained in the Abstract and Introduction, even though it is the key mechanism to be discussed in the review. I had no idea about the concept until reading section 4. This needs to be explained earlier.

We greatly appreciate the reviewer's feedback and have modified the abstract and the introduction to reiterate the sponge effect concept. We hope that this clarification of the sponge effect model better prepares the reader for the discussion in section 4.

Abstract:

TFs are the master regulators of cellular identity, capable of driving cell fate transitions including differentiations, reprogramming and transdifferentiations. Pioneer TFs recognize partial motifs exposed on nucleosomal DNA, allowing for TF-mediated activation of repressed chromatin. Moreover, there is evidence suggesting that certain TFs can repress actively expressed genes either directly through interactions with accessible regulatory elements or indirectly through mechanisms that impact the expression, activity or localization of other regulatory factors. Recent evidence suggests that during reprogramming, the reprogramming TFs initiate opening of chromatin regions rich in somatic TF motifs that are inaccessible in the initial and final cellular states. We postulate that analogous to a sponge, these transiently accessible regions "soak up" somatic TFs, hence lowering the initial barriers to cell fate changes. This indirect TF-mediated event, which we've aptly named the "sponge effect", may play an essential role in the silencing of the somatic transcriptional network.

Introduction:

In this perspective, we discuss the nature of pioneer TFs, the proposed mechanistic models behind direct and indirect TF-mediated cellular identity conversions and how these mechanistic aspects could be exploited to further enhance in vitro cell fate changes. Furthermore, we discuss how TF-mediated decondensation of motif-rich "sponge" chromatin can directly sequester TFs away from gene regulatory elements (REs) to expedite the extinction of established TF networks. We theorise that through this process, which we coined the "sponge effect", cell fate altering TFs initiate redistribution of TFs away from active REs to promote silencing of the initial cell identity network.

2. It is unclear whether sponge-like regions only function in redistributing somatic TFs in chromatin or also function as cis-regulatory regions for activating target genes during cellular reprogramming.

We agree with the reviewer that one of the key questions regarding the sponge regions is whether they also (at least to some extent) act as regulatory elements. This in turn would mean that TF relocation to these sites has additional functional consequences apart from silencing of the initial cell identity network (e.g. activation of transiently expressed genes).

Indeed, our analysis of MEF reprogramming (Cell Stem Cell 2017) revealed that opening of these transiently accessible regions and expression of the “closest” genes correlate positively at the later stages of reprogramming. However, at the beginning of reprogramming, transient opening of these OCT4/SOX2 target sites does not correlate with transcriptional activation of the closest genes but with a decrease in expression of MEF identity genes and closure of MEF regulatory elements. This suggests that many of these transiently accessible chromatin regions indeed function as cis-regulatory elements towards the end of reprogramming when the cells transition towards pluripotency. However, the majority of these transiently accessible regions and OCT4/SOX2 target sites do not seem to act as cis-regulatory elements at the beginning of reprogramming or during the loss of somatic cell identity. Nevertheless, it is possible that these regions (at least some) act as distal regulatory elements as we acknowledge that the closest gene may not necessarily be the gene that is regulated. In order to address this interesting point raised by the reviewer, we have now added the following sentences for further clarification of the proposed sponge effect.

*However, the majority of OCT4 and SOX2 binding events observed were to regions inaccessible in MEFs and iPSCs. (Knaupp et al., 2017a) **Notably, transient opening of these chromatin regions does not generally correlate with transcriptional activation of the closest genes. Instead, opening of these regions coincides with silencing of the MEF identity network. This suggests that the majority of OCT4 and SOX2 target sites at the start of reprogramming are not cis-REs and most likely have different functions. Whether some of these transient regions act as distal REs remains largely unknown. Interestingly, these transiently accessible chromatin regions are not only enriched with the motifs of OCT4, SOX2 and KLF4 but also various somatic TF motifs including members of the AP-1, TEAD, RUNX and ETS TF families. (Knaupp et al., 2017a; Li et al., 2017a) Therefore, we proposed that OCT4, SOX2 and KLF4 engage with closed chromatin regions to open them transiently. This in turn initiates the redistribution of somatic factors to starve fibroblast identity genes of TFs, indirectly forcing the extinction of the somatic transcriptional network (Figure 3C).***

Editor: When revising your manuscript, please aim for concise wording wherever possible. In agreement with the reviewers, I felt that this Perspective was overall already quite well-written, but I do think there are some places where you may be able to streamline the wording. Please also keep an eye out for long paragraphs and consider breaking them into shorter paragraphs. I think the Introduction, for example, would be a good place where this may be helpful.

We would like to thank you for accepting our manuscript for publication in Advanced Genetics. We have addressed all reviewer comments and made several modifications to the original manuscript - highlighted in attached PDF. We have also broken the introduction into smaller paragraphs. We hope you find these changes satisfactory. Please see a summary of minor edits below:

Blue highlights indicate where a paragraph was split into two.

The following sentence in the introduction:

- TFs usually engage other regulatory proteins including other TFs, histone modifiers and chromatin remodelers to exert their functions.

Was changed to:

- To exert their function, TFs usually engage other regulatory proteins, including other TFs, histones modifiers and chromatin remodelers.

Insight into the **underlying** fundamental—mechanisms through which TFs mediate changes in cell identity is therefore key to our understanding of human development and tumorigenesis. This **basic—knowledge** **fundamental information** is of major relevance.....

References

- Chanda, S., Ang, C. E., Davila, J., Pak, C., Mall, M., Lee, Q. Y., Ahlenius, H., Jung, S. W., Südhof, T. C., & Wernig, M. (2014). Generation of induced neuronal cells by the single reprogramming factor ASCL1. *Stem Cell Reports*, 3(2), 282–296.
- Chen, J., Chen, X., Li, M., Liu, X., Gao, Y., Kou, X., Zhao, Y., Zheng, W., Zhang, X., Huo, Y., Chen, C., Wu, Y., Wang, H., Jiang, C., & Gao, S. (2016). Hierarchical Oct4 Binding in Concert with Primed Epigenetic Rearrangements during Somatic Cell Reprogramming. *Cell Reports*, 14(6), 1540–1554.
- Chen, X., Xu, H., Yuan, P., Fang, F., Huss, M., Vega, V. B., Wong, E., Orlov, Y. L., Zhang, W., Jiang, J., Loh, Y.-H., Yeo, H. C., Yeo, Z. X., Narang, V., Govindarajan, K. R., Leong, B., Shahab, A., Ruan, Y., Bourque, G., ... Ng, H.-H. (2008). Integration of external signaling pathways with the core transcriptional network in embryonic stem cells. *Cell*, 133(6), 1106–1117.
- Chronis, C., Fiziev, P., Papp, B., Butz, S., Bonora, G., Sabri, S., Ernst, J., & Plath, K. (2017). Cooperative Binding of Transcription Factors Orchestrates Reprogramming. *Cell*, 168(3), 442–459.e20.
- Davis, R. L., Weintraub, H., & Lassar, A. B. (1987). Expression of a single transfected cDNA converts fibroblasts to myoblasts. *Cell*, 51(6), 987–1000.
- Du, Y., Wang, J., Jia, J., Song, N., Xiang, C., Xu, J., Hou, Z., Su, X., Liu, B., Jiang, T., Zhao, D., Sun, Y., Shu, J., Guo, Q., Yin, M., Sun, D., Lu, S., Shi, Y., &

- Deng, H. (2014). Human hepatocytes with drug metabolic function induced from fibroblasts by lineage reprogramming. *Cell Stem Cell*, 14(3), 394–403.
- Gurdon, J. B. (1962). The developmental capacity of nuclei taken from intestinal epithelium cells of feeding tadpoles. *Journal of Embryology and Experimental Morphology*, 10, 622–640.
- Hussein, S. M. I., Puri, M. C., Tonge, P. D., Benevento, M., Corso, A. J., Clancy, J. L., Mosbergen, R., Li, M., Lee, D.-S., Cloonan, N., Wood, D. L. A., Munoz, J., Middleton, R., Korn, O., Patel, H. R., White, C. A., Shin, J.-Y., Gauthier, M. E., Lê Cao, K.-A., ... Nagy, A. (2014). Genome-wide characterization of the routes to pluripotency. *Nature*, 516(7530), 198–206.
- Ieda M. (2011). [Direct reprogramming of fibroblasts into functional cardiomyocytes by defined factors]. *Nihon rinsho. Japanese journal of clinical medicine*, 69 Suppl 7, 524–527.
- Knaupp, A. S., Buckberry, S., Pflueger, J., Lim, S. M., Ford, E., Larcombe, M. R., Rossello, F., de Mendoza, A., Alaei, S., Firas, J., Holmes, M. L., Nair, S. S., Clark, S. J., Nefzger, C. M., Lister, R., & Polo, J. M. (2017a). Transient and Permanent Reconfiguration of Chromatin and Transcription Factor Occupancy Drive Reprogramming. *Cell Stem Cell*, 21(6), 834–845.e6.
- Knaupp, A. S., Buckberry, S., Pflueger, J., Lim, S. M., Ford, E., Larcombe, M. R., Rossello, F., de Mendoza, A., Alaei, S., Firas, J., Holmes, M. L., Nair, S. S., Clark, S. J., Nefzger, C. M., Lister, R., & Polo, J. M. (2017b). Transient and Permanent Reconfiguration of Chromatin and Transcription Factor Occupancy Drive Reprogramming. *Cell Stem Cell*, 21(6), 834–845.e6.
- Li, D., Liu, J., Yang, X., Zhou, C., Guo, J., Wu, C., Qin, Y., Guo, L., He, J., Yu, S., Liu, H., Wang, X., Wu, F., Kuang, J., Hutchins, A. P., Chen, J., & Pei, D. (2017a). Chromatin Accessibility Dynamics during iPSC Reprogramming. *Cell Stem Cell*, 21(6), 819–833.e6.
- Li, D., Liu, J., Yang, X., Zhou, C., Guo, J., Wu, C., Qin, Y., Guo, L., He, J., Yu, S., Liu, H., Wang, X., Wu, F., Kuang, J., Hutchins, A. P., Chen, J., & Pei, D. (2017b). Chromatin Accessibility Dynamics during iPSC Reprogramming. *Cell Stem Cell*, 21(6), 819–833.e6.
- Liu, X., Ouyang, J. F., Rossello, F. J., Tan, J. P., Davidson, K. C., Valdes, D. S., Schröder, J., Sun, Y. B. Y., Chen, J., Knaupp, A. S., Sun, G., Chy, H. S., Huang, Z., Pflueger, J., Firas, J., Tano, V., Buckberry, S., Paynter, J. M., Larcombe, M. R., ... Polo, J. M. (2020). Reprogramming roadmap reveals route to human induced trophoblast stem cells. *Nature*, 586(7827), 101–107.
- Polo, J. M., Anderssen, E., Walsh, R. M., Schwarz, B. A., Nefzger, C. M., Lim, S. M., Borkent, M., Apostolou, E., Alaei, S., Cloutier, J., Bar-Nur, O., Cheloufi, S., Stadtfeld, M., Figueroa, M. E., Robinton, D., Natesan, S., Melnick, A., Zhu, J., Ramaswamy, S., & Hochedlinger, K. (2012). A molecular roadmap of reprogramming somatic cells into iPS cells. *Cell*, 151(7), 1617–1632.

- Roberts, G. A., Ozkan, B., Gachulincová, I., O'Dwyer, M. R., Hall-Ponsele, E., Saxena, M., Robinson, P. J., & Soufi, A. (2021). Dissecting OCT4 defines the role of nucleosome binding in pluripotency. *Nature Cell Biology*, 23(8), 834–845.
- Schwarz, B. A., Cetinbas, M., Clement, K., Walsh, R. M., Cheloufi, S., Gu, H., Langkabel, J., Kamiya, A., Schorle, H., Meissner, A., Sadreyev, R. I., & Hochedlinger, K. (2018). Prospective Isolation of Poised iPSC Intermediates Reveals Principles of Cellular Reprogramming. *Cell Stem Cell*, 23(2), 289–305.e5.
- Sen, R., & Baltimore, D. (1986). Inducibility of kappa immunoglobulin enhancer-binding protein Nf-kappa B by a posttranslational mechanism. *Cell*, 47(6), 921–928.
- Sen, R., & Baltimore, D. (2006). Multiple nuclear factors interact with the immunoglobulin enhancer sequences. *Cell* 1986. 46: 705-716. *Journal of Immunology*, 177(11), 7485–7496.
- Soufi, A., Donahue, G., & Zaret, K. S. (2012). Facilitators and impediments of the pluripotency reprogramming factors' initial engagement with the genome. *Cell*, 151(5), 994–1004.
- Soufi, A., Garcia, M. F., Jaroszewicz, A., Osman, N., Pellegrini, M., & Zaret, K. S. (2015). Pioneer transcription factors target partial DNA motifs on nucleosomes to initiate reprogramming. *Cell*, 161(3), 555–568.
- Sridharan, R., Tchieu, J., Mason, M. J., Yachechko, R., Kuoy, E., Horvath, S., Zhou, Q., & Plath, K. (2009). Role of the murine reprogramming factors in the induction of pluripotency. *Cell*, 136(2), 364–377.
- Vierbuchen, T., Ostermeier, A., Pang, Z. P., Kokubu, Y., Südhof, T. C., & Wernig, M. (2010). Direct conversion of fibroblasts to functional neurons by defined factors. *Nature*, 463(7284), 1035–1041.
- Yamamizu, K., Piao, Y., Sharov, A. A., Zsiros, V., Yu, H., Nakazawa, K., Schlessinger, D., & Ko, M. S. H. (2013). Identification of transcription factors for lineage-specific ESC differentiation. *Stem Cell Reports*, 1(6), 545–559.
- Zhang, Y., Pak, C., Han, Y., Ahlenius, H., Zhang, Z., Chanda, S., Marro, S., Patzke, C., Acuna, C., Covy, J., Xu, W., Yang, N., Danko, T., Chen, L., Wernig, M., & Südhof, T. C. (2013). Rapid single-step induction of functional neurons from human pluripotent stem cells. *Neuron*, 78(5), 785–798.
- Zhuang, Q., Li, W., Benda, C., Huang, Z., Ahmed, T., Liu, P., Guo, X., Ibañez, D. P., Luo, Z., Zhang, M., Abdul, M. M., Yang, Z., Yang, J., Huang, Y., Zhang, H., Huang, D., Zhou, J., Zhong, X., Zhu, X., ... Esteban, M. A. (2018). NCoR/SMRT co-repressors cooperate with c-MYC to create an epigenetic barrier to somatic cell reprogramming. *Nature Cell Biology*, 20(4), 400–412.

Dear Dr Knaupp,

Thank you for submitting your revised manuscript entitled "Indirect mechanisms of transcription factor-mediated gene regulation during cell fate changes" (Perspective, No. ggn2.202200015R1) to Advanced Genetics. I am happy to write that we are now satisfied with the content and are glad to accept this Perspective, in principle, for publication at the journal. I have only two rather minor comments, listed below, which I would ask you to consider:

1. While reading through this Perspective again, I was reminded of a paper from 2012 by Lee and Maheshri (<https://doi.org/10.1038/msb.2012.7>). While this work was performed in yeast and did not discuss cell fate regulation in metazoans, it was one of the first papers (to my knowledge) to quantitatively explore the effects of decoy TF binding sites on gene regulation. Their conclusion that decoy binding sites can transform regulation to a bimodal regime also seems relevant to your proposal here. Please consider whether it might be relevant to cite and briefly comment on this work in your paper.

2. Please ensure that TF is defined on first use in your abstract.

After considering and addressing these two final points, please submit a final version with a brief cover letter. This version will be checked and then sent rapidly to our production team for final copy editing and publication

To submit your revision, go to <https://www.editorialmanager.com/advgenet/> and log in as an Author using your username (*****) and password. Your submission can be found under the menu item "Submissions Needing Revision". The changes to your manuscript should be highlighted in a different color in the primary "Revised Manuscript" file.

We should receive your revised manuscript by 22 Sep 2022. Once we receive your revised manuscript, we will provide a final decision as soon as possible.

Yours sincerely,

Andrew Hufton

P.S. Please help avoid delays by referring to the Manuscript Preparation Checklist (<http://www.advgenet.com/authorguidelines>) and use the appropriate article template when preparing your revised manuscript. Please also follow the instructions to prepare and upload your Production Data materials. These include: the full, non-highlighted text of your manuscript (with all figures with a resolution of at least 300 dpi, schemes, and tables) in editable format - Word DOC/X or LaTeX; a short summary (50-60 words) and an eye-catching color image for the Journal's Table of Contents; and, if applicable, Supporting Information*.

--

Dr Andrew Hufton, Editor
Advanced Genetics
E-mail: AdvGenet@wiley.com
Tel: +49(0)6201-606-362

<http://www.advgenet.com>

Authors' Response to 2nd Review

03 Oct 2022

Dear Andrew,

Thank you for reviewing and accepting our manuscript. We have addressed the two final comments and surmised the changes below.

- 1) While reading through this Perspective again, I was reminded of a paper from 2012 by Lee and Meheshri (<https://doi.org/10.1038/msb.2012.7>). While this work was performed in yeast and did not discuss cell fate regulation in metazoans, it was one of the first papers (to my knowledge) to quantitatively explore the affects of decoy TF binding sites on gene regulation. Their conclusion that decoy binding sites can transform regulation to a bimodal regime also seems relevant to your proposal here. Please consider whether it might be relevant to cite and briefly comment on this work in your paper.

We agree that the 2012 publication by Lee and Maheshri is a fitting inclusion for the discussion in chapter five and would like to thank you for this suggestion. Should these binding dynamics discovered in their rTA-tetO system be extended to transcription factors binding to repetitive or motif-rich chromatin in higher organisms, sequestration may serve as a mechanism to finetune or desensitize regulatory responses including those that drive cell fate decisions. We have adjusted the manuscript as outlined below.

expression, reducing the accumulation of excess matrix components like collagen and

limited the severity of fibrosis in rats with tubulointerstitial fibrosis.^[153]

Altogether, these reports are in agreement with the idea that sequestration

of TFs by newly available binding sequences can effectively manipulate

existing gene expression networks. Interestingly, a study using a synthetic

system in budding yeast suggests that decoy sites may not simply serve as

competitive inhibitors but can also transform the regulatory graded dose-

response of a positive-feedback loop into a bimodal (switchlike)

response.[1] The authors suggest that this provides a simple mechanism

to generate an ultrasensitive regulatory response that is necessary for dynamic biological networks.[1]

Whether dynamically activated sponge-like heterochromatin exists as an integral component of an already complicated gene regulatory framework remains to be determined, however, this provides a fascinating new alternative to traditional direct transcriptional regulation.

- 2) We have updated the abstract to include the TF definition “*transcription factor (TF)*”.

References:

[1] “A regulatory role for repeated decoy transcription factor binding sites in target gene expression” - Tek-Hyung Lee and Narendra Maheshri. *Molecular systems biology* 2012.

Final Decision	04 Oct 2022
----------------	-------------

Dear Dr Knaupp,

Thank you for submitting your manuscript entitled "Indirect mechanisms of transcription factor-mediated gene regulation during cell fate changes" (Perspective, No. ggn2.202200015R2) to *Advanced Genetics*.

I'm pleased to inform you that your manuscript has been accepted for publication without change.

We will copyedit the accepted version of your manuscript and if we require any further information at this stage we will contact you. After copyediting we will let you know when you can expect to receive the proofs. Instructions for returning your proof corrections will be provided when the proofs are sent to you.

Please note, as a matter of course, if you have used content from other sources, it is your responsibility to request permission to reproduce or adapt the content. This may apply to your own previous publications. If you have not obtained permission to reproduce any content yet, please do so immediately.

All articles published in *Advanced Genetics* are fully open access: immediately and freely available to read, download and share. *Advanced Genetics* charges a publication fee to cover publication costs. The corresponding author for this manuscript should have already received a quote with the article publication fee, and will soon receive an e-mail invitation to register with or log in to Wiley Author Services (<https://authorservices.wiley.com>). After logging into Wiley Author Services, the publication fee

can be paid by credit card, or an invoice or pro forma can be requested. Payment of the publication charge must be received before the article will be published online.

Congratulations on your results, and thank you for choosing Advanced Genetics for publishing your work. I hope you will consider us for the publication of your future manuscripts.

Yours sincerely,

Andrew Lee Hufton, PhD

P.S.: If you believe your images might be appropriate for use on the cover of Advanced Genetics, and you would like your paper to be considered for the cover, please e-mail us your layout suggestions with a short description. For details on cover image preparation, please see the cover gallery on <http://www.advgenet.com>.

--

Dr Andrew Hufton, Editor
Advanced Genetics
E-mail: AdvGenet@wiley.com
Tel: +49(0)6201-606-362

<http://www.advgenet.com>